# NeuZip: Memory-Efficient Training and Inference with Dynamic Compression of Neural Networks

## Abstract

The performance of neural networks improves when more parameters are used. However, the model sizes are constrained by the available on-device memory during training and inference. Although applying techniques like quantization can alleviate the constraint, they suffer from performance degradation. In this work, we introduce NeuZip, a new weight compression scheme based on the entropy of floating-point numbers in neural networks. With NeuZip, we are able to achieve memory-efficient training and inference without sacrificing performance. Notably, we significantly reduce the memory footprint of training a Llama-3 8B model from 31GB to less than 16GB, while keeping the training dynamics fully unchanged. In inference, our method can reduce memory usage by more than half while maintaining near-lossless performance.

## 1 Introduction

Deep learning with neural networks has become the backbone of numerous artificial intelligence applications. The search for better-performing networks is a longstanding topic in deep learning. Without modifying the design, scaling up the number of parameters (e.g., number of hidden dimensions or layers) has been demonstrated as an effective practice to boost the performance of neural networks of the same kind (Kaplan et al., 2020). This idea has been successfully applied to text, image, audio, and multi-modal tasks with a wide range of model architectures (Yu et al., 2022; Radford et al., 2019; Brown et al., 2020). Recently, the number of parameters in the state-of-the-art models has become more than 100 billion or even a trillion parameters. For example, one of the state-of-the-art language models in 2020, GPT-3, has 175B parameters (Brown et al., 2020), growing by nearly 100 times compared with the largest Transformer architecture in the 2017 paper (Vaswani et al., 2017).

Despite the growth in the model size, the hardware capacity is not keeping up with the pace: the largest on-device memory of GPUs was 32GB in 2017, and is 80GB to this date in 2024, growing by only 2.5 times. The available hardware supply poses a limitation on the trainable model size, bottlenecking the scaling capacity. Although this problem can be alleviated by using more GPUs and sharding the model in multiple devices (Rajbhandari et al., 2019), such a practice introduces more communication overheads among GPUs, making large-scale distributed training less efficient. Therefore, saving the total memory usage is critical in scaling up neural networks.

The peak memory usage is dominated by three relatively independent parts: the optimizer, the saved activations for back-propagation, and the model itself. For the optimizer, there are already memory-efficient optimizers achieving a sublinear space complexity (Shazeer & Stern, 2018; Hao et al., 2024); for the activations, the memory can be saved by enabling activation checkpointing (Chen et al., 2016), which saves the storage by recomputing the forward activations during the back-propagation. For the model parameters, there has not been an effective method to save the memory while preserving the ability to train the model. Recently, Dettmers et al. (2023) proposed the quantized low-rank adaptation (QLoRA), which freezes the parameters using a 4-bit data type for the backbone pre-trained model. While significantly saving the memory for the model, it imposes a constraint on the overall change of the model to be low-rank, limiting the model capacity.

In this paper, we propose NeuZip, an algorithm to compress the neural networks while maintaining their full abilities. Specifically, each floating-point number is represented by three parts: the sign bit, the exponent bits, and the mantissa bits. Following the observation that weights are concentrated around zero (Kalamkar et al., 2019), we demonstrate that this corresponds to the low-entropy nature of the exponent bits. We hence compress the exponent bits using the asymmetric numeral system (ANS Duda (2013)), a lossless compression algorithm that achieves a high throughput on parallel computing devices like GPUs. Since the compression is lossless, the memory reduction comes without compromising any precision loss and enables full-parameter training.

In addition to lossless compression for training, we also propose a lossy variant of NeuZip for inference that further reduces the memory footprint. Specifically, we control the relative change of each parameter by storing only the top-$k$ significant bits of the mantissa. We empirically show that lossy NeuZip lies at the Pareto frontier of the memory–performance trade-off when compared with several state-of-the-art quantization baselines.

## 2 OUR APPROACH

The Shannon entropy (Shannon, 1948) is used to measure the "stochasticity" of a random variable with the following definition:

$$H(X) := \mathop{\mathbb{E}}_{X \sim p(X)} \left[ -\log_2 p(X) \right] \tag{1}$$

for a random variable $X$ with probability $p$. A lower entropy indicates a less stochasticity of a random variable. In fact, the entropy equals the minimum number of bits required, in expectation, to represent a random variable, therefore corresponding to data compressibility. For the non-concentrating random variable with all possible values sharing an equal probability, the entropy of which reaches the maximum value $\log_2 n$, where $n$ is all possible values $X$ can take. On the other hand, for highly-concentrating (e.g., fully deterministic) random variables, the entropy can be as low as 0.

### 2.1 LOW-ENTROPY NATURE OF NEURAL NETWORK PARAMETERS

We argue that the parameters in neural network tend to have low entropy. First, parameters are typically initialized with Gaussian distribution for matrices (Glorot & Bengio, 2010; He et al., 2015). This encourages all weights to be centered around zero, effectively reducing the entropy (or randomness). In addition, regularization is also applied for better generalization ability. For example, the weight decay technique reduces the magnitudes of weights at every update iteration. Similarly in Bayesian inference, prior distributions (e.g., Gaussian and Laplace distributions) are often applied, imposing a zero-concentrated preference over the parameters. Even without explicit regularization, stochastic gradient descent (SGD) or its variants are shown to have the implicit regularization effect on neural networks, meaning the model parameters are implicitly encouraged to have smaller magnitudes during training (Soudry et al., 2018; Vardi & Shamir, 2021). All the above effects and techniques lead to the following observation:

**Observation 2.1** *Assuming neural network parameters are i.i.d. random variables, the entropy of the distribution is likely to be low.*

Specifically, each parameter is represented is represented by three components: the sign bit, the exponent bits, and the mantissa bits in the IEEE 754 standard (IEEE, 2019).[1] Therefore, we conduct a fine-grained analysis and investigate the distribution of each component of a floating-point number in neural networks.

As shown in Figure 1, the sign bit has a high entropy as it is evenly distributed; hence, it is not compressible. For the exponent bits, there is a clear pattern that they demonstrate a low-entropy nature, carrying only less than 3 bits of information with 8 bits of capacity. For the mantissa bits, they store nearly 7-bit information with 7-bit capacity. In fact, we shown in Appendix A that this is common in deep learning.

---

[1]We use BF16 (Kalamkar et al., 2019) in this paper.

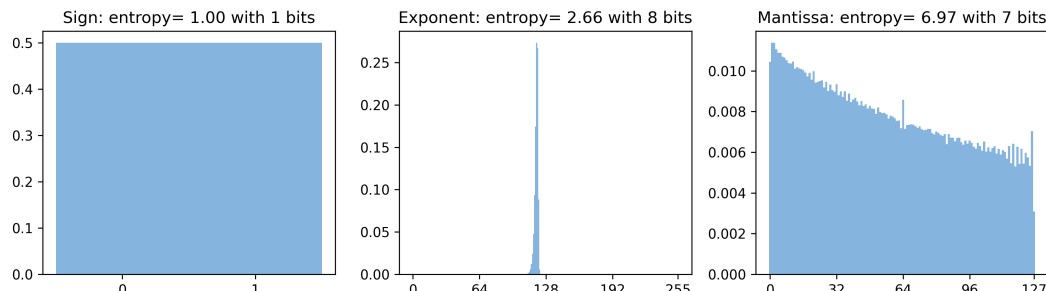

Figure 1: The histograms of different components of the parameters of LLama-3 8B model (Dubey et al., 2024). The $x$-axis is all possible binary values and the $y$-axis represent the frequency of each value.

This phenomenon suggests that by simply compressing the exponents, we are able to recover the overall optimal compression ratio. In this example, an ideal compression algorithm is able to achieve a ratio as high as 1.501 (the sum of the three entropy values), only marginally below the overall compression ratio 1.505.

## 2.2 LOSSLESS NEUZIP: COMPRESSING EXPONENTS FOR TRAINING

**Compressed representation.** Based on observation, we see that the number of bits per exponent is largely inflated compared with the information entropy. However, previous research demonstrates that the dynamic range provided by the 8-bit exponents are critical for neural networks (Kalamkar et al., 2019). We therefore propose to compress the exponent bits in a lossless manner based on the entropy. This practice mainly has three benefits: (1) it increases the throughput of compression as only part of the bits are processed by the compression; (2) it reduces the burden of maintaining the statistics of a large set of symbols (e.g., 256 symbols for 8-bit exponents versus 65,536 symbols for 16-bit representations), enabling a great efficiency of compression algorithms; (3) most importantly, it recovers most of the compressibility as shown in Figure 1.

**Multi-layer neural networks.** The compression alone does not save any memory for maintaining a single array. This is because, either compression or decompression, requires at least one buffer of the same size as the uncompressed array. In the scope of neural networks, the whole model is prohibitively large and it is infeasible to duplicate the memory. In NeuZip, however, we exploit the multi-layer structure of modern neural networks to avoid creating a large buffer. Without loss of generality (see Appendix F), we focus on the linear function as a common building block in neural networks at layer $l$:

$$\boldsymbol{x}_l \leftarrow \boldsymbol{W}_l \boldsymbol{x}_{l-1} + \boldsymbol{b}_l, \tag{2}$$

where $\boldsymbol{W}_l \in \mathbb{R}^{m \times n}$ is the weight matrix, $\boldsymbol{b}_l \in \mathbb{R}^m$ is the bias vector of layer $l$, and $\boldsymbol{x}_l$ is the input of layer $l$. We propose to modify the compressed forward pass in the following form

$$\hat{\boldsymbol{W}} \leftarrow \text{decompress}(\boldsymbol{c}_l) \tag{3}$$

$$\boldsymbol{x}_l \leftarrow \hat{\boldsymbol{W}} \boldsymbol{x}_{l-1} + \boldsymbol{b}_l, \tag{4}$$

where $\boldsymbol{c}_l$ is the compressed storage of the matrix $\boldsymbol{W}_l$. In this way, we only need to store $\boldsymbol{c}_i$ for each layer, enjoying low-memory usage. During each forward pass, weight matrices stay in the compressed form until the original data is needed, in which case it is decompressed into a temporary space $\hat{\boldsymbol{W}}$ for computation. As a result, the entire network is never fully decompressed at any point in time, making the overall forward pass memory efficient. The per-layer procedure is shown in Figure 2.

Note that although we alter the forward pass, the back-propagation for each linear layer is fully unaffected. This is because

$$\frac{\partial \mathcal{L}}{\partial \boldsymbol{W}_l} = \frac{\partial \mathcal{L}}{\partial \boldsymbol{x}_l} \frac{\partial \boldsymbol{x}_l}{\partial \boldsymbol{W}_l} = (\nabla_{\boldsymbol{x}_l} \mathcal{L}) \boldsymbol{x}_{l-1}^\top. \tag{5}$$

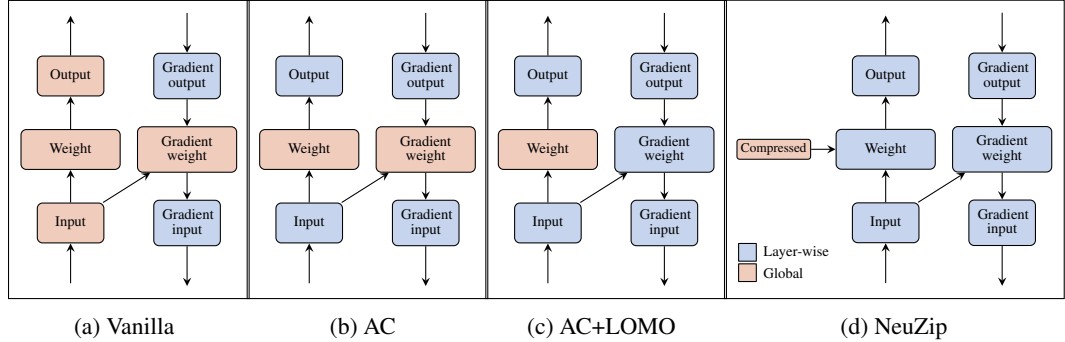

(a) Vanilla      (b) AC      (c) AC+LOMO      (d) NeuZip

Figure 2: Reverse-mode automatic differentiation (e.g., back-propagation) with different memory-saving techniques for a linear layer. Blocks colored blue are loaded in memory temporarily for the calculation of this layer, whereas the blocks colored red are always in memory throughout training.

Therefore, we are able to obtain the gradient as long as the activations are saved. Similarly, we can also propagate the gradient of inputs with

$$\frac{\partial \mathcal{L}}{\partial \boldsymbol{x}_{l-1}} = \frac{\partial \mathcal{L}}{\partial \boldsymbol{x}_l}\frac{\partial \boldsymbol{x}_l}{\partial \boldsymbol{x}_{l-1}} = (\nabla_{\boldsymbol{x}_l}\mathcal{L})^\top \boldsymbol{W}_l, \tag{6}$$

where $\boldsymbol{W}_l$ can be constructed by decompression. It is worth noting that our NeuZip is compatible with activation checkpointing (Chen et al., 2016) by recomputing the activations, opening more opportunity for memory saving.

For weight updates, we decompress the matrix into the original floating-point format and compress the updated matrix again. This procedure is done in a layer-by-layer fashion, similar to LOMO (Lv et al., 2023). The overall training procedure is described in Appendix B.

**Compression algorithm.** In our implementation, we choose to use the asymmetric numeral systems (ANS) (Duda, 2013) as our backbone compression algorithm because it can be easily parallelized and achieves a high throughput with parallel execution, making it an ideal candidate on deep learning accelerators like GPUs. Specifically, ANS encodes a sequence of symbols by treating them as base-$n$ numbers. However, unlike the common numerical system that uses a uniform base for each digit, ANS treats every single digit with a different base $\lceil 1/\hat{p}_i \rceil$, where $\hat{\boldsymbol{p}}$ is the frequency of symbols. As a result, it achieves a near-optimal compression rate by suing around $1/\hat{p}_i$ bits for the $i^{\text{th}}$ symbol.

## 2.3 Lossy NeuZip: Additionally Truncating Mantissa for Inference

In the algorithm above, we show that the training of neural networks can be completely unaffected by lossless compression. On the other hand, inference is known to be less sensitive to precision loss compared with training (Dettmers et al., 2022; Dettmers & Zettlemoyer, 2023). This enables further memory reduction of NeuZip by reducing the precision. In our study, we conduct a pilot experiment that perturbs each weight with a noise proportional to the weight magnitude. We observe that with a small noise ratio there is little or no effect on the overall performance (Appendix C). Motivated by this, we propose a variant of NeuZip that compresses mantissa in a lossy way during inference.

In its core, we simply round and truncate the mantissa to fewer bits. Specifically, we assume the original floating-point number $f$ has an exponent $e$ and mantissa $m$. After rounding, the mantissa is denoted by $\hat{m}$ and the resulting floating-point number is denoted by $\hat{f}$.

The rounding introduces an error expressed as:

$$|f - \hat{f}| = \left| 2^{e-127} \cdot \frac{m}{2^7} - 2^{e-127} \cdot \frac{\hat{m}}{2^7} \right| = 2^{e-134} \cdot |m - \hat{m}| \tag{7}$$

where $e - 127$ interprets the exponent bits $e$ as an integer, which can be either positive, negative, or 0. In the fraction $m/2^7$, $m$ is the significand (an unsigned integer) and 7 is the precision. It is

straightforward to see that the relative error is given by

$$\frac{|f - \hat{f}|}{|f|} = \frac{2^{e-134} \cdot |m - \hat{m}|}{2^{e-134} \cdot |m|} = \frac{|m - \hat{m}|}{m}. \tag{8}$$

Suppose our rounding keeps $k$ most significant bits in $\hat{m}$, the earliest point where $\hat{m}$ could differ from the original number $m$ is at the $(k+1)$th bit. This means that the maximum possible relative change introduced by this rounding is $1/2^k$. Given that the mantissa bits are highly uniform as shown in Figure 1, such a practice resembles the weight perturbation based on relative magnitudes, justifying the rounding trick applied to mantissas.

In our implementation, we store the sign and mantissa bits together as a signed integer to minimize the requests of memory write. Further, given that modern architectures are mostly byte (8-bit) addressable, we pack multiple such signed integers into a single byte for memory efficiency. To align with an 8-bit byte, we let the precision after rounding to be $\{0, 1, 3\}$, ensuring that all the bits in a byte are utilized efficiently. We illustrate the process in Figure 8b.

Lastly, we enable a block-wise normalization technique (Dettmers et al., 2023), where a block is a chunk of weights that are stored contiguously in memory. Such block-wise normalization makes sure that the weight with the largest magnitude in a block will always be normalized to 1, invariant to mantissa rounding and truncation. The normalization coefficient—which handles mantissa while ignoring the exponent—is stored with 8 bits, and is used for de-normalization during the decompression of the weight. This strategy is based on the observation that larger weights play a more important role in neural networks (Han et al., 2015).

## 3 EXPERIMENTS

We empirically verify the effectiveness of NeuZip across different model architectures and datasets. Given the success of large language models, we mainly consider Transformer-based models for our experiments. We choose two designs of Transformer, decoder-only and encoder–decoder models, to show the generality of our method. Each experiment is conducted on a single RTX A6000 GPU to avoid complications with the memory-usage in the multi-gpu scenario. The uncompressed data type is set to BF16 for all experiments.

### 3.1 LOSSLESS NEUZIP FOR PRE-TRAINING

**Settings.** We choose decoder-only models to evaluate our method on the pre-training task. We select 3 models with different sizes to study the scaling effect, including GPT-Neo 2.7B (Black et al., 2021), Llama-3 8B (Dubey et al., 2024), and LLama-2 13B (Touvron et al., 2023). For fair comparison, all competing methods are initialized with the same random weights.

For the task, we consider language modeling, which requires the model to predict the next token given the context. We use the Wikitext-2 dataset (Merity et al., 2016), where each data sample is a fixed-length sequence from an article on Wikipedia. We set the length to 1024 following the common practice (Radford et al., 2019).

For each experiment, we report the loss (negative log-likelihood) on unseen samples. To study memory saving, we report the peak memory usage for each run during the training process. The numbers are shown in gibibyte (GiB, $1024^3$ bytes). We also report the speed by the number of iterations per second to demonstrate the time-efficiency of each method.

We apply the vanilla SGD update to all runs for efficiency. The activation checkpointing technique (Chen et al., 2016) is enabled by default. It is worth noting that pre-training these large models to the optimal performance is extremely expensive (Rajbhandari et al., 2019). Given that our NeuZip training method is lossless, we only train the models for 1 epoch to showcase its effectiveness. We use the same hyper-parameters for all runs.

**Results.** We present the results in Table 1. We first test the vanilla training method, where only the activation checkpointing is applied (shown in Figure 2b). As shown, the vanilla training requires the highest amount of memory because it stores the uncompressed weights and gradients for all layers.

Table 1: Pre-training decoder-only models on the language modeling task. The loss numbers are calculated on the validation set with the cross-entropy loss. Memory is reported in GiB ($1024^3$ B). Speed represents the number of iterations per second. The **bold** numbers represent the top results.

| Name | GPT-Neo-XL 2.7B | | | Llama-3 8B | | | LLama-2 13B | | |
|---|---|---|---|---|---|---|---|---|---|
| | Loss | Mem | Speed | Loss | Mem | Speed | Loss | Mem | Speed |
| Vanilla | **8.81** | 11.22 | **0.96** | **8.61** | 30.97 | 0.77 | - | OOM | - |
| LOMO | **8.81** | 6.97 | 0.94 | **8.61** | 19.47 | **0.78** | **9.10** | 26.26 | **0.49** |
| +NeuZip Lossless | **8.81** | **5.54** | 0.70 | **8.61** | **15.25** | 0.45 | **9.10** | 18.58 | 0.28 |

Table 2: Fine-tuning encoder–decoder models on the SQL generation task. The BLEU scores are calculated with SacreBLEU. Memory is reported in GiB ($1024^3$ B). Speed represents the number of iterations per second. The **bold** numbers represent the top results.

| Name | T5 1B | | | T5 3B | | | T5 11B | | |
|---|---|---|---|---|---|---|---|---|---|
| | BLEU | Mem | Speed | BLEU | Mem | Speed | BLEU | Mem | Speed |
| Vanilla | **79.9** | 3.82 | **3.69** | **85.1** | 11.32 | 2.43 | - | OOM | - |
| LOMO | **79.9** | 2.75 | 3.68 | **85.1** | 7.07 | **2.47** | **82.3** | 25.95 | **0.69** |
| + NeuZip Lossless | **79.9** | **2.39** | 2.02 | **85.1** | **5.21** | 1.33 | **82.3** | **20.68** | 0.46 |
| QLoRA INT8 | 70.4 | 5.84 | 1.11 | 72.1 | 11.54 | 1.12 | 63.5 | 33.36 | 0.37 |
| QLoRA FP4 | 70.1 | 3.63 | 1.70 | 72.1 | 7.35 | 1.74 | 63.3 | 22.73 | 0.58 |
| QLoRA FP4$^2$ | 70.6 | 3.61 | 1.63 | 72.0 | 7.27 | 1.61 | 60.6 | 22.38 | 0.57 |
| QLoRA NF4 | 70.4 | 3.63 | 1.83 | 71.2 | 7.35 | 1.65 | 59.4 | 22.73 | 0.57 |
| QLoRA NF4$^2$ | 70.5 | 3.61 | 1.64 | 71.2 | 7.07 | 1.57 | 57.9 | 22.38 | 0.57 |

We also test the LOMO technique (Lv et al., 2023), which promptly updates the weights in a layer-by-layer fashion (shown in Figure 2c). This allows LOMO to reuse a buffer to store the gradients for each layer. As a result, LOMO approximately reduces the peak memory usage by the size of a model.

Finally, we apply our NeuZip on top of LOMO (shown in Figure 2d). For all models, NeuZip additionally reduces more than 20% percentage of memory compared with LOMO, accounting for a total memory reduction of more than 50%. Notably, NeuZip reduces the peak memory of training a Llama-2 13B model to less than 20GB, enabling training a 13B model on consumer-grade GPUs without any precision loss.

## 3.2 Lossless NeuZip for Fine-Tuning

**Settings.** A benefit of using lossless compression comes from retaining the pre-trained weight without any information loss. We conduct a fine-tuning experiment with encoder–decoder models to test the performance of our NeuZip on broader architectures. In particular, we choose three T5 models: T5 1B, T5 3B, and T5 11B (Raffel et al., 2020), where the pre-trained parameters are used for initialization.

The T5 models are pre-trained on the C4 dataset (Lin et al., 2020), which is filtered to contain natural language only. To avoid data leaks from pre-training, we choose a non-natural language generation dataset for fine-tuning. Specifically, we use a public SQL generation dataset (Zhong et al., 2017; Yu et al., 2018) as the test bed. For each sample, the model is required to generate the SQL command from a human question. For example, the question could be "`CREATE TABLE head (age INTEGER)`. How many heads of the departments are older than 56 ?". The model is expected to generate "`SELECT COUNT(*) FROM head WHERE age > 56`". We feed the question and response into the encoder and decoder, respectively. The objective is to minimize the cross-entropy loss on the response.

Table 3: Evaluating lossy NeuZip on different models and tasks. 'PPL" represents the perplexity values. Memory is reported in GiB. Speed represents the number of iterations per second. The **bold** numbers represent the top results, whereas the underlined numbers are the second-best ones.

(a) Evaluating decoder-only models on the language modeling task. Here, the perplexities are adjusted to word level to compare across different tokenizations.

| Name | Llama-3 8B | | | Llama-2 13B | | | Yi-1.5 34B | | |
|---|---|---|---|---|---|---|---|---|---|
| | PPL | Mem | Speed | PPL | Mem | Speed | PPL | Mem | Speed |
| Vanilla | **9.89** | 15.08 | **5.07** | **10.87** | 24.36 | **3.59** | - | OOM | - |
| Quant INT8 | 10.07 | 8.63 | 3.54 | 10.97 | 12.74 | 2.27 | 10.87 | 33.41 | 1.13 |
| Quant FP4 | 11.51 | 5.77 | 3.45 | 11.38 | 7.37 | 1.87 | 11.57 | 19.54 | **1.75** |
| Quant NF4 | 10.75 | 5.77 | 3.38 | 11.15 | 7.37 | 1.83 | 11.06 | 19.54 | 1.67 |
| Quant FP4$^2$ | 11.50 | 5.44 | 3.41 | 11.38 | 6.87 | 1.86 | 11.57 | 18.11 | 1.61 |
| Quant NF4$^2$ | 10.75 | 5.44 | 3.34 | 11.15 | 6.87 | 1.81 | 11.06 | 18.11 | 1.54 |
| NeuZip 0-bit | 13.64 | **5.24** | 3.44 | 12.46 | **6.30** | 1.87 | 12.06 | **16.20** | 0.94 |
| NeuZip 1-bit | 10.77 | 6.05 | 3.38 | 11.17 | 7.77 | 1.86 | 11.04 | 20.14 | 0.93 |
| NeuZip 3-bit | 9.93 | 7.70 | 3.38 | 10.90 | 10.73 | 1.84 | 10.76 | 27.92 | 0.93 |
| NeuZip 7-bit (lossless) | **9.89** | 10.95 | 3.39 | **10.87** | 16.66 | 1.84 | 10.72 | 43.40 | 0.94 |

(b) Evaluating encoder–decoder models on the language modeling task. Since all models use the same tokenizer, we reported perplexities at the token level for simplicity.

| Name | T5 1B | | | T5 3B | | | T5 11B | | |
|---|---|---|---|---|---|---|---|---|---|
| | PPL | Mem | Speed | PPL | Mem | Speed | PPL | Mem | Speed |
| Vanilla | **2.614** | 1.37 | **23.73** | **2.571** | 5.31 | **19.86** | **2.568** | 21.06 | **6.20** |
| Quant INT8 | 2.615 | 1.28 | 4.24 | 2.573 | 4.94 | 4.28 | 2.569 | 19.59 | 2.58 |
| Quant NF4 | 2.632 | 1.08 | 11.64 | 2.588 | 4.12 | 11.82 | 2.579 | 16.28 | 4.48 |
| Quant FP4 | 2.646 | 1.08 | 11.92 | 2.594 | 4.12 | 11.99 | 2.585 | 16.28 | 4.59 |
| Quant FP4$^2$ | 2.646 | 1.05 | 10.39 | 2.594 | 4.03 | 9.72 | 2.585 | 15.93 | 4.52 |
| Quant NF4$^2$ | 2.632 | 1.05 | 10.39 | 2.587 | 4.03 | 9.96 | 2.579 | 15.93 | 4.39 |
| NeuZip 0-bit | 2.731 | **0.40** | 11.82 | 2.668 | **1.41** | 8.70 | 2.651 | **5.35** | 3.24 |
| NeuZip 1-bit | 2.641 | 0.48 | 11.68 | 2.591 | 1.78 | 8.61 | 2.581 | 6.65 | 3.21 |
| NeuZip 3-bit | **2.614** | 0.66 | 11.99 | 2.574 | 2.42 | 8.60 | 2.569 | 9.27 | 3.19 |
| NeuZip 7-bit (lossless) | **2.614** | 0.99 | 11.55 | **2.571** | 3.73 | 8.77 | **2.568** | 14.46 | 3.23 |

Similar to the pre-training experiments, we also sweep the learning rate from $10^{-3}$ to $3 \times 10^{-1}$ for each run. After fine-tuning, we generate with the model on the validation set with greedy decoding. The generated SQL commands are then compared with the ground truths by SacreBLEU (Post, 2018), a metric that evaluates the similarity between corpora based on precision scores.

**Results.** The results are reported in Table 2. All baselines in the pre-training experiment (i.e., the vanilla training, LOMO, and NeuZip) are included in this table. Similar to the results in Section 3.1, they achieve the same BLEU scores for each model. Specifically, our NeuZip is able to train a 11B model within 24GB.

For fine-tuning, it is possible to apply other memory-efficient training techniques. For example, QLoRA (Dettmers et al., 2023) compresses the pre-trained model by using low-precision data types and train the LoRA modules only (Hu et al., 2022). In our comparison experiment, we choose the widely used quantization data types for QLoRA, including INT8 (Dettmers et al., 2022), FP4, and NF4 (Dettmers et al., 2023). We apply the LoRA modules (Hu et al., 2022) on all linear layers, where every LoRA rank is set to $8$ to control the memory usage.[2] As shown in the second half of Table 2, all quantization methods underperform NeuZip in terms of both generation quality and

---

[2]It should be noted that the down-projection matrices in each T5 feed-forward network are not quantized for stability, as otherwise the model performance is seriously jeopardized. See `https://github.com/huggingface/transformers/issues/20287` for more details.

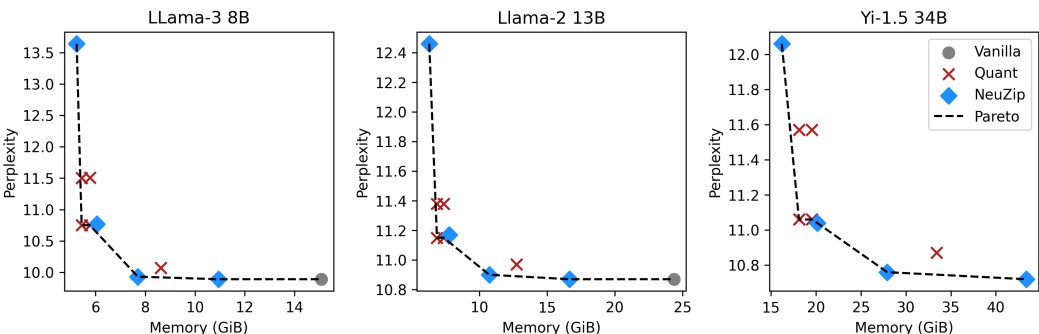

Figure 3: The trade-off between memory and performance for different methods.

memory usage. In terms of time efficiency, some quantization methods are slower than others, but in general, they are in the same magnitude as our method. Overall, NeuZip achieves the least memory usage while maintaining the highest performance. The results strongly suggests the practicality of our NeuZip.

### 3.3 Lossy Compression for Inference

As mentioned in Section 2.3, the inference process is less sensitive in precision loss, which provides an opportunity for compressing mantissa in a lossy fashion during inference. We evaluate the performance of our lossy NeuZip in such scenarios.

**Settings.** Following the settings in previous sections, we test our approach with both decoder-only and encoder–decoder architectures. For the decoder-only models, we select the LLama-3 8B (Dubey et al., 2024), LLama-2 13B (Touvron et al., 2023), and Yi-1.5 34B (Young et al., 2024). For the encoder–decoder architecture, we use the T5 1B, 3B, and 11B models as in Section 3.2.

Since all decoder-only models are trained for language modeling, we evaluate the performance with language modeling tasks. Specifically, we test all methods on the Wikitext-2 validation set (Merity et al., 2016) following Section 3.1, where each sequence consists of 1024 tokens. On the other hand, the encoder–decoder models (T5 series) contain multiple tasks in pre-training. Since they excel at zero-shot translation, we evaluate them on the WMT14 En-De translation task (Bojar et al., 2014), where each source sentence is prepended with "translate from English to German:" based on the pre-training format (Raffel et al., 2020).

Following the standard evaluation pipeline for lossy compression (Frantar et al., 2023; Dettmers & Zettlemoyer, 2023), we evaluate all models with the perplexity metric, which is sensitive to how distorted the compressed model is.

**Results.** The results for decoder-only and encoder–decoder models are shown in Tables 3a and 3b, respectively. We see that the vanilla (uncompressed BFloat16) models achieve the best perplexity scores in all experiments at a cost of the excessive memory usage. For quantization methods, we choose the same INT8 (Dettmers et al., 2022), FP4, and NF4 (Dettmers et al., 2023) data types mentioned in Section 3.2. In general, quantization methods suffer from notable perplexity degradation. Although the INT8 variant (Dettmers et al., 2022) manages to better preserve the perplexity, it uses around 50% more memory compared with other quantization methods.

For our lossy NeuZip, we set three different levels of precision: 0-bit, 1-bit, and 3-bit mantissa preserved. We choose these values because they are aligned in 8-bit byte arrays (discussed in Section 2.3). All these variants use a block size of 512 for normalization. We additionally include the lossless NeuZip (7-bit mantissa) for a full comparison. As shown in the table, our lossy NeuZip demonstrates a spectrum of memory saving and performance preservation. The 0-bit NeuZip attains the best memory efficiency in all experiments, whereas the lossless 7-bit NeuZip obtains the best perplexity scores. Notably, the 3-bit NeuZip achieves nearly lossless performance in all experiments while using less than 50% memory compared with the uncompressed model. The results confirm the effectiveness of our method.

Table 4: The effect of block size.

| Name | Block 32 | | Block 64 | | Block 128 | | Block 256 | | Block 512 | |
|---|---|---|---|---|---|---|---|---|---|---|
| | PPL | Mem | PPL | Mem | PPL | Mem | PPL | Mem | PPL | Mem |
| NeuZip 0-bit | 6.341 | 35.7 | 6.694 | 34.6 | 6.853 | 34.2 | 7.639 | 33.8 | 7.104 | 33.5 |
| NeuZip 1-bit | - | OOM | 4.611 | 42.7 | 4.662 | 42.2 | 4.640 | 41.8 | 4.649 | 41.4 |

## 3.4 IN-DEPTH ANALYSES

**The memory–performance trade-off.** In Section 3.3, we observe that the performance is generally decreased with less memory usage. We analyze this trade-off of our NeuZip as well as quantization methods in Figure 3. Note that the optimal methods are the ones on the Pareto frontier (Pareto, 2014), i.e., the more bottom-left, the better. In addition to measuring the perplexity, we also include a preliminary study by evaluating the end-to-end performance on the MMLU dataset (Hendrycks et al., 2020) in Appendix E.

As shown, three out of four NeuZip variants are on the Pareto frontier, with the remaining one staying fairly close to the frontier. On the other hand, there is only one quantization technique that lies on the Pareto frontier. This result demonstrates that our NeuZip generally achieves a better memory–performance trade-off than quantization.

**The effect of block size in lossy compression.** As introduced in Section 2, we apply normalization to lossy NeuZip to ensure the weight with the largest absolute value will not be affected by truncation. We show the effect of block size in this experiment with a giant model, Llama-3 70B evaluated on the Wikitext-2 dataset.

As seen in Table 4, a smaller block size clearly leads to better performance at the cost of compromising memory efficiency due to the overhead of storing normalization coefficients. Therefore, the block-wise normalization provides a more fine-grained trade-off between memory and performance by varying the block size.

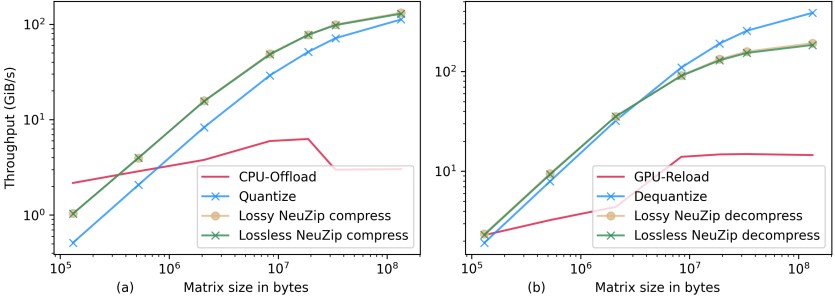

Figure 4: The throughput experiment. (a) Comparison of CPU-offloading, quantization, lossy NeuZip compression, and lossless NeuZip compression. (b) Comparison of GPU-reloading, de-quantization, lossy NeuZip decompression, and lossless NeuZip decompression.

**Throughputs of NeuZip.** In addition to overall time efficiency presented in Tables 1–3, we analyze the throughput of matrix compression and decompression with our NeuZip, in comparison with the throughput of matrix quantization and de-quantization based on the NF4 data type (Dettmers et al., 2023) using the popular library `bitsandbytes`.[3] We additionally include the CPU-offloading technique as a baseline, which lowers the GPU memory pressure by transferring data to CPU and reloading them to GPU when needed. Figure 4 measures the throughput of matrix processing in GiB/s when we vary the matrix size from $10^5$ to $10^8$ bytes.

We see that CPU-offloading is generally slow across different sizes of matrices. This is due to the bottleneck of CPU–GPU communication through PCIe. For quantization, the `bitsandbytes`

---

[3]Available at `https://github.com/bitsandbytes-foundation/bitsandbytes`

package has been highly optimized for GPU, and its throughput is one magnitude higher than the CPU-offloading technique when the matrix size is large. Profoundly, our NeuZip achieves the highest throughput for compression among all methods (Figure 4a), and a high throughput for decompression similar to de-quantization (Figure 4b). The results suggest that our NeuZip, albeit causing overhead compared with uncompressed vanilla models, is still highly efficient in practice.

## 4 RELATED WORK

**Model compression.** Previous work has explored different techniques to reduce the memory usage of neural networks, including knowledge distillation (Hinton et al., 2015) and pruning (Kwon et al., 2022). Most related to our work is the quantization technique, which represents each parameter with fewer bits; common approaches include $k$-means-based quantization (Han et al., 2016), linear quantization (Han et al., 2016), mixed precision quantization (Dettmers et al., 2022; 2023), and non-uniform grid quantization (Chikin & Antiukh, 2022). Our NeuZip differs from these methods by utilizing the exponent entropy and achieving lossless compression (see Appendix G for details). When training data are available, one may incorporate the quantization into the training process to improve performance (Xiao et al., 2023; Frantar et al., 2023). In this paper, our NeuZip compression is a zero-shot method, and therefore, our experiments consider the widely used zero-shot quantization methods (Dettmers et al., 2022; 2023) for fair comparison. We leave the utilization of additional data of NeuZip to future work.

**Memory-efficient optimizers.** The optimizer also occupies a considerable amount of memory during training (Rajbhandari et al., 2019). To address this, memory-efficient optimizers (Shazeer & Stern, 2018; Zhao et al., 2024; Hao et al., 2024) are developed to reduce the memory footprint of training. Our NeuZip is orthogonal to these optimization techniques, as it can be seamlessly combined with any of these methods for further memory saving. In particular, the lossless NeuZip is expected to have exactly the same results with less memory.

**Parameter-efficient training.** Another line of research saves memory by training a subset of parameters (Houlsby et al., 2019; Zaken et al., 2022) so the optimizer only stores information about a small set of trainable parameters. One notable example is the low-rank adaptation (LoRA (Hu et al., 2022)). However, such a practice restricts the optimization space of parameters, and thus usually leads to significant performance degradation. Moreover, low-rank methods are unsuitable for pre-training.

It is important to mention that memory-efficient optimizers and parameter-efficient training cannot reduce the memory cost during inference. By contrast, our NeuZip is suitable for both training and inference.

## 5 CONCLUSION

**Summary.** In this work, we present NeuZip, a novel compression scheme for neural networks that achieves memory-efficient training and inference. By analyzing the floating-point structures, we propose to compress the exponent in a lossless way and to compress the mantissa in a lossy way. The lossless variant of our NeuZip may be applied to both training and inference, while yielding exactly the same result as the uncompressed model. The lossy NeuZip provides additional memory saving for inference, achieving superior memory–performance trade-off.

**Limitations and future work.** Due to the hardware constraint, the largest model that we consider in this paper is 70B. We would like to verify our NeuZip on even larger models like GPT-3 (Brown et al., 2020) with more capable hardware. Another limitation of NeuZip is that the throughput is lower than the vanilla model. However, it has a comparable speed with highly optimized quantization methods while achieving significantly better performance. By using NeuZip, we expect to create opportunities for researchers to explore and study large models on more accessible or edge devices.

## ETHICS STATEMENT

Our research focuses on accelerating pre-trained Transformers to reduce power and time consumption in inference. As far as we are aware, it does not pose any new ethical or societal risks that require specific attention.

## REPRODUCIBILITY STATEMENT

In our paper, all of our models and datasets are publicly accessible. For the models, we download them from HuggingFace's hub using the following identifiers (ranked from small to large):

- `google-t5/t5-large`
- `EleutherAI/gpt-neo-2.7B`
- `google-t5/t5-3b`
- `meta-llama/Meta-Llama-3-8B`
- `google-t5/t5-11b`
- `meta-llama/Llama-2-13b-hf`
- `01-ai/Yi-1.5-34B`
- `meta-llama/Meta-Llama-3-70B`
- `state-spaces/mamba-2.8b-hf`

For the datasets, we follow the previous work and similarly obtain them from HuggingFace:

- `Salesforce/wikitext`
- `b-mc2/sql-create-context`
- `wmt/wmt14`

For the procedure of NeuZip, we have detailed descriptions in Section 2 and also with Algorithm 1. We plan to release our code at this anonymous link.

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

## A    INSPECTING THE ENTROPY ON MORE MODELS

**Random initialization.**    When training from scratch, the parameters are randomly initialized. To verify the compressibility in this case, we check the parameter entropy of a randomly initialized model with the same architecture as Lllama-3. The initialization methods follow the standard procedure provided in the Hugging Face library. The results show a similar pattern to what the released Llama model has, suggesting the compressibility with NeuZip occurs even with random weights.

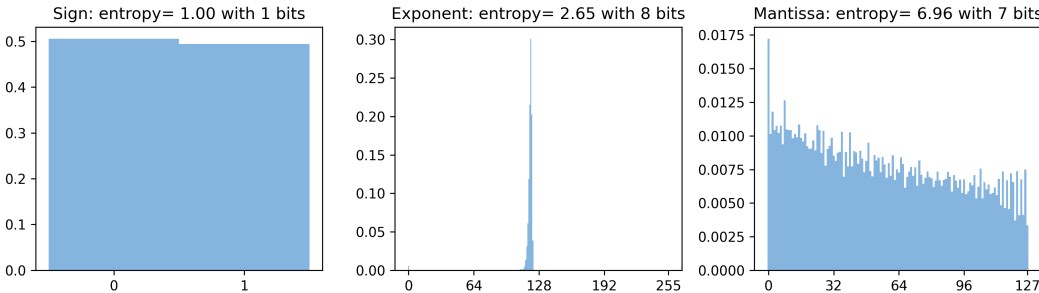

Figure 5: The histograms of different floating-point components of the parameters of a randomly initialized Llama-3 8B model.

**Diffusion.**    We also inspect the parameter entropies beyond Transformer models. In Figure 6, we check all four models in a diffusion pipeline. We see that the low-entropy exponents not only occur in Transformer models but other architectures like convolution-based VAE and U-Net models.

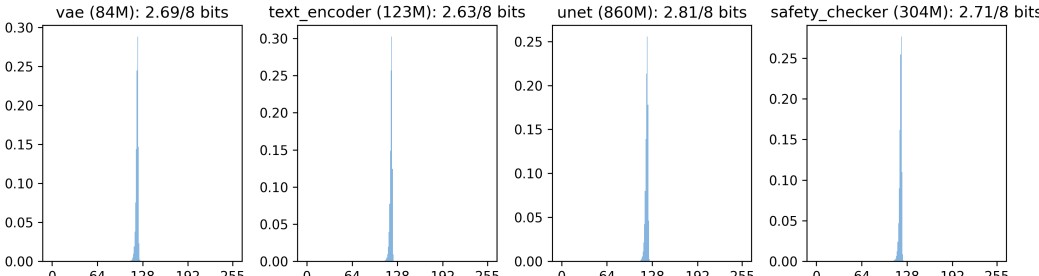

Figure 6: The histograms of the exponent bits in different components of Stable Diffusion 1.5 model. We omit the sign and mantissa bits for simplicity as we do not compress them based on their entropies.

Both experiments show that the occurrence of low-entropy components is a common phenomenon in deep learning.

## B    THE ALGORITHM FOR TRAINING WITH LOSSLESS NEUZIP

In this section, we describe the forward-backward procedure of NeuZip. First, we compress all the linear layers in the original model and store the compressed information on-device. During the training iterations, we decompress the compressed weights in a layer-by-layer manner for the forward pass. For the backward pass, the input is recalculated again following the forward pass like activation checkpointing (Chen et al., 2016). A linear operation calculates the gradients for both the weight matrix and input. To do so, we need to decompress the weight again, which is used to calculate the gradient of input. After the gradient is calculated, we directly update the weight without storing it similar to LOMO (Lv et al., 2023).

---

**Algorithm 1** Memory-efficient training with NeuZip

---

**Require:** number of linear layers $L$, linear layer weights $\{\boldsymbol{W}_i\}_{i=1}^{L}$.
**Require:** data stream $\mathcal{D}$ that yields training data $\boldsymbol{x}$ for each iteration
    ▷ Initialization
1: **for** $l \in 1 \ldots L$ **do**
2:    $\boldsymbol{s}_l, \boldsymbol{e}_l, \boldsymbol{m}_l \leftarrow \text{split}(\boldsymbol{W}_l)$        ▷ Split each element in the matrix into three components
3:    $\boldsymbol{c}_l \leftarrow \text{compression}(\boldsymbol{e}_l)$        ▷ Compress the exponents losslessly
4:    $\text{store}(\boldsymbol{s}_l, \boldsymbol{c}_l, \boldsymbol{m}_l)$        ▷ Store the compressed exponents $\boldsymbol{c}_L$ on device
5: **end for**
    ▷ Training loop
6: **for** $\boldsymbol{x}$ in $\mathcal{D}$ **do**
7:    ▷ Model forward
8:    $\boldsymbol{x}_0 \leftarrow \boldsymbol{x}$
9:    **for** $l \in 1 \ldots L$ **do**
10:     $\hat{\boldsymbol{e}} \leftarrow \text{decompression}(\boldsymbol{c}_l)$        ▷ Decompress the exponents using temporary space
11:     $\hat{\boldsymbol{W}} \leftarrow \text{merge}(\boldsymbol{s}_l, \hat{\boldsymbol{e}}_l, \boldsymbol{m}_l)$ ▷ Concatenate into a floating-point number matrix using temporary space

12:     $\boldsymbol{x}_l \leftarrow \hat{\boldsymbol{W}}^{\top}\boldsymbol{x}_{l-1} + \boldsymbol{b}_l$        ▷ Linear calculation
13:     $\text{save\_for\_backward}(\boldsymbol{x}_l)$        ▷ Label the variable required for back-propagation
14:    **end for**
15:    ▷ Model backward and update
16:    $\Delta_{\boldsymbol{x}} \leftarrow \partial\mathcal{L}/\partial\boldsymbol{x}_L$        ▷ Calculate the gradient w.r.t. the model output
17:    **for** $l \in L \ldots 1$ **do**
18:     $\hat{\boldsymbol{e}} \leftarrow \text{decompression}(\boldsymbol{c}_l)$        ▷ Decompress the exponents using temporary space
19:     $\hat{\boldsymbol{W}} \leftarrow \text{merge}(\boldsymbol{s}_l, \hat{\boldsymbol{e}}_l, \boldsymbol{m}_l)$ ▷ Concatenate into a floating-point number matrix using temporary space

20:     $\Delta_{\boldsymbol{W}} \leftarrow (\Delta_{\boldsymbol{x}})\boldsymbol{x}_{l-1}^{\top}$        ▷ Calculate gradient by Equation (5)
21:     $\hat{\boldsymbol{W}} \leftarrow \hat{\boldsymbol{W}} - \text{optimizer}(\Delta_{\boldsymbol{W}})$        ▷ Update the weight on-the-fly
22:     $\boldsymbol{s}_l, \boldsymbol{e}_l, \boldsymbol{m}_l \leftarrow \text{split}(\hat{\boldsymbol{W}})$        ▷ Split each element in the matrix into three components again
23:     $\boldsymbol{c}_l \leftarrow \text{compression}(\boldsymbol{e}_i)$        ▷ Compress the exponents losslessly again
24:     $\text{store}(\boldsymbol{s}_l, \boldsymbol{c}_l, \boldsymbol{m}_l)$        ▷ Replace the stored components for layer $l$ on device
25:     $\Delta_{\boldsymbol{x}} \leftarrow \hat{\boldsymbol{W}}\Delta_{\boldsymbol{x}}$        ▷ Calculate the gradient of input for next layers
26:    **end for**
27: **end for**

---

## C  THE TOLERANCE OF RANDOM PERTURBATION

In this experiment, we would like to explore the sensitivity of neural network weights to random perturbations. For each parameter, we have two types of magnitudes: absolute and relative magnitudes. The former one represents the actual numerical error, whereas the second one is calculated based on the original value. For example, when the original value is $-1.5$, an absolute magnitude of $0.125$ means the perturbed range is $[-1.5 - 0.125, -1.5 + 0.125]$. On the other hand, a relative magnitude of $0.125$ means the perturbed range is $[-1.5*(1+0.125), -1.5*(1-0.125)]$. We conduct such a experiment with the perturbation grid in Figure 7. For each cell, we choose the maximum error between relative error and absolute value for perturbing. A random value is sampled from the perturbed range uniformly as the perturbation. The weight value is then set to the random sample.

As shown in Figure 7, we see a clear pattern that the model tends to tolerate the relative change rather than the absolute change.

## D  THE STORAGE FOR LOSSLESS AND LOSSY COMPRESSION

In this section, we describe the underlying storage layout for NeuZip in Figure 8.

Essentially, each BFloat16 number is first split into an exponent and a signed mantissa. We group all the exponents in the matrix and perform the lossless compression. The signed mantissa is optionally truncated, depending on the required precision. The signed mantissa is then stored separately in memory.

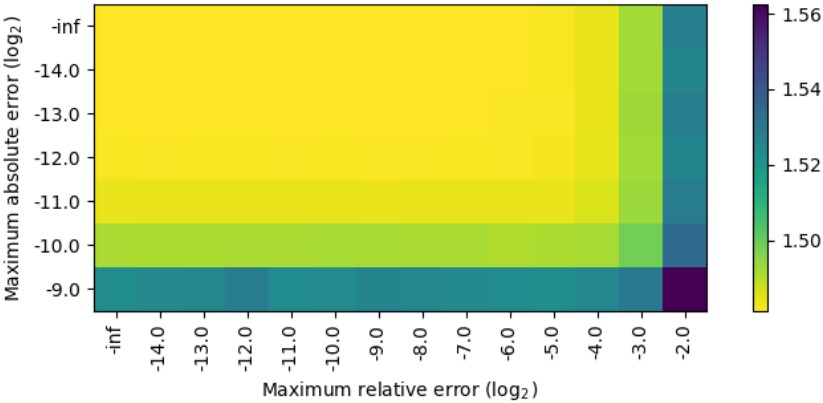

Figure 7: Evaluating the byte-level perplexity with perturbed LLama-3 8B model (Dubey et al., 2024) on Wikitext-2 (Merity et al., 2016). Each parameter is perturbed with controlled noises. Both the $x$- and $y$-axes are log-scale with base 2.

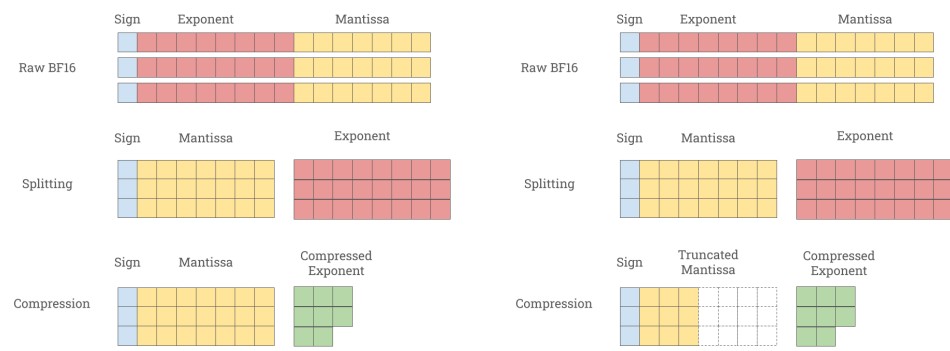

(a) Lossless compression scheme.
(b) Lossy compression scheme (with 3 bits).

Figure 8: The storage structures for NeuZip.

# E    EVALUATING ON MMLU

We provide the results on MMLU (Hendrycks et al., 2020) in Figure 9. Here, the theoretical optimal point should be at the top left corner.

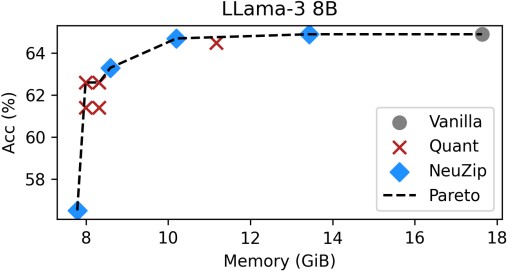

Figure 9: The memory–performance trade-off on the MMLU dataset.

Similar to the results in Section 3.4, all of our NeuZip variants are on the Pareto frontier, suggesting the optimal trade-off between memory and performance.

# F  BEYOND LINEAR LAYERS

**Generalization to advanced architectures.**  Given the design of Transformer models, we mainly compress linear layers and present the corresponding back-propagation algorithm during training. However, it is straightforward that NeuZip can be applied to broader architectures. For example, we can compress the convolution layers or recurrent units in the same way. This is because both operations involve the linear operation as the basic component (e.g. convolution can be viewed as multiple local linear transformations). The forward and backward passes follow the same analyses presented in Section 2.2. We further demonstrate this by evaluating a state space model, Mamba 2.8B (Gu & Dao, 2023). Mamba model is an ideal testbed because it uses convolution layers with the recurrence feature in inference. The test results on Wikitext-2 (Merity et al., 2016) are shown in Table 5.

Table 5: Evaluating NeuZip with Mamba 2.8B on Wikitext-2. 'PPL' represents the word-level perplexity values. Memory is reported in GiB. Speed represents the number of iterations per second.

|  | PPL | Memory | Speed |
|---|---|---|---|
| Vanilla | 16.54 | 5.23 | 0.190 |
| Quant FP4 | 17.52 (+5.9%) | 1.68 (-67.9%) | 0.189 (-0.5%) |
| Quant NF4 | 17.12 (+3.5%) | 1.68 (-67.9%) | 0.193 (+1.6%) |
| Quant FP4$^2$ | 17.55 (+6.1%) | 1.57 (-70.0%) | 0.185 (-2.6%) |
| Quant NF4$^2$ | 17.15 (+3.7%) | 1.57 (-70.0%) | 0.186 (-2.1%) |
| NeuZip 0-bit | 17.26 (+4.4%) | 1.43 (-72.7%) | 0.184 (-3.2%) |
| NeuZip 1-bit | 17.26 (+4.4%) | 1.73 (-66.9%) | 0.188 (-1.1%) |
| NeuZip 3-bit | 16.61 (+0.4%) | 2.34 (-55.3%) | 0.186 (-2.1%) |
| NeuZip 7-bit (lossless) | 16.54 (+0.0%) | 3.62 (-30.8%) | 0.189 (-0.5%) |

As displayed, lossless NeuZip saves around 30% of the memory usage without compromising the perplexity and inference speed. Moreover, lossy NeuZip like the 3-bit version saves more than half of the memory while only marginally decreases the perplexity (by only 0.4%). The results not only shows the versatility of NeuZip on a wide range of model architectures but also indicate a strong ability in compressing the memory.

**Non-linear activation functions.**  In some implementations, more intermediate activation values are saved to accelerate back-propagation. For example, the Gaussian error linear unit (GELU, Hendrycks & Gimpel (2016)) activation function may save the exponential values of the inputs for gradient calculation.[4] In these cases, the memory saving percentage will be slightly lower for parameter compression methods like quantization and NeuZip. However, we can apply activation checkpointing (Chen et al., 2016) during training for more memory saving by recomputing the additional activations.

# G  DISCUSSION ON QUANTIZATION AND COMPRESSION

Our NeuZip resembles quantization in the sense that both methods save the parameter memory. However, their mechanisms for memory saving are fundamentally different. Specifically, quantization is inherently lossy regardless of the chosen quantization values. The introduced error in this lossy process hinders its applicability in scenarios where precision plays an important role. As a concrete example, quantization-based training methods underperform NeuZip in both quality and memory saving in Section 3, displaying the importance of precisions.

---

[4]This might only hold for certain implementations.

Undoubtedly, some quantization schemes may have lower levels of precision loss. For example, if the quantized values are uniformly distributed (i.e. linear quantization Han et al. (2015); Dettmers et al. (2022)), the overall precision loss could be enormous because it overlooks the distribution of the raw data. On the other hand, using non-uniform quantization could alleviate this issue by providing tighter approximations for common values (Chikin & Antiukh, 2022; Dettmers et al., 2023). However, even non-uniform schemes cannot entirely eliminate precision loss and often rely on sub-optimal mixed-precision approaches to compensate for the reductions (Dettmers et al., 2022; 2023). More importantly, recent studies have highlighted that precision becomes increasingly critical in large-scale settings (Kumar et al., 2024). This observation reinforces the importance of applying NeuZip, which enables precision control up to a lossless level.

It is important to note that NeuZip is specifically designed for the raw parameters of neural networks and is not practical for application to quantized values. This is because optimal quantization schemes yield (approximately) equal frequency distributions across all quantized values (Chikin & Antiukh, 2022; Dettmers et al., 2023). As a result, applying additional compression would neither improve the compression rate nor preserve efficiency, instead halving the throughput.

