# OpenReview forum: "NeuZip: Memory-Efficient Training and Inference with Dynamic Compression of Neural Networks"
_ICLR.cc/2025/Conference — Submitted to ICLR 2025_

### Official Review · Reviewer_DiBQ · 2024-10-21

**Soundness:** 2
**Presentation:** 2
**Contribution:** 1
**Rating:** 3
**Confidence:** 4

**Summary:**

The paper presents NeuZip a compression technique for reducing the memory requirements of neural networks for training and inference. NeuZip contains both a lossless and a lossy operating mode, where either only the exponent bits are (losslessly) compressed, or additionally a truncation of the mantissa bits is performed. The performance is tested on various models from the Natural Language Processing community.

**Strengths:**

- The paper is well structured, the experiments are well described and the figures and tables are informative.
- The developed method in the paper is well explained and clear to understand.
- The experiment settings and presentation of results are well chosen to show the effect of proposed method.

**Weaknesses:**

- The novelty of the _NeuZip_ method seems poor.  Lossless NeuZip consists of the direct application of a very widely used compression method (ANS) to (parts of) the weight vector. The lossy compression part is a truncation of mantissa bits, which is a well-known method in data compression (f.e. used in the GRIB2 format). Moreover, the paper fails to mention that this truncation itself is equivalent to quantisation to a non-uniform grid (f.e. float32 with 0 mantissa bits corresponds to quantization with the grid $\{\pm 2^{-127}, \pm2^{-126}, \dots \pm2^{128}\}$). Quantization with non-uniform grids is a widely used idea (f.e. in "Data Free Network Quantization", Chikin et al. 2022, or NF4 from QLoRA, Dettmers et al. 2023).
- The methods that the paper compares lossless _NeuZip_ to are not fairly chosen, and this is not immediately made obvious in the experimental descriptions. In Table 2, the quantisation methods that _NeuZip_ compares to contain no entropy coding, while _NeuZip_ contains entropy-coding in the exponent bits (which will be especially significant as the ratio of exponent bits to mantissa bits increases). Including entropy coding into the compared quantisation methods would reduce the speed by a small bit, but increase the compression performance at no perplexity cost (which are the metrics the methods are compared on). Again, as the proposed method is equivalent to non-uniform grid quantisation, I would expect most of the gains that are reported in this table to have come from the included entropy coding.
- A large part of the memory efficiency gains of lossless _NeuZip_ come from the established LOMO method (see Table 1). Adding _NeuZip_ on top often reduces the execution speed by a large margin.

**Questions:**

- I wonder how the reported speeds for the quantization methods such as Quant INT8 in Table 2 came to be. Especially for large models, inference speed-ups should be expected, see Dettmers et al. 2023 (LLM.int8(), https://arxiv.org/abs/2208.07339), appendix D, p.17ff, where LLM.int8 reports increased inference speed for all models that have more than 13B parameters).
- Similarly, for QLoRA, which has been shown to make networks more memory efficient in the original work, this paper seems to report contrary results, decreasing or barely reducing the memory efficiency compared to vanilla networks as well as suffering a large performance hit (f.e. T5 1B). A discussion of why this happens could be beneficial to the paper, and if it turns out that QLoRA is not suitable for networks this small, other state-of-the-art methods should rather be used to compare _NeuZip_ with.

---

> ### Author Response · Authors · 2024-11-20
> **Official Comment by Authors (1/2)**
>
> We thank the reviewer for providing the comments.
>
> > Weakness 1 (novelty of exponent compression): “The novelty of the NeuZip method seems poor. Lossless NeuZip consists of the direct application of a very widely used compression method (ANS) to (parts of) the weight vector.”
>
> Our contribution lies in identifying the proper components to compress and is agnostic to the underlying compression algorithm. In fact, without our observation (shown in Figure 1), the entropy encoding scheme is not practical because even 16-bit representations will require 65,535 additional statistics without a clear entropy pattern. By contrast, we only need to maintain the 255 statistics for 8-bit representations with a clear low-entropy pattern (discussed in lines 133-137). Hence, we believe that our work makes a novel observation and proposes a novel yet effective approach (although not a new compression algorithm), given that the exponent entropy was not exploited before.
>
> > Weakness 1 (novelty of mantissa truncation): “The lossy compression part is a truncation of mantissa bits, which is a well-known method in data compression (f.e. used in the GRIB2 format).”
>
> Mantissa truncation is derived from our observation that neural networks are not sensitive to random perturbations in the relative scale (as opposed to the absolute scale) shown in Figure 7, allowing us to further simplify the relative perturbation into mantissa truncation by revealing their connections in Section 2.3. This connection is novel and has not been sufficiently explored before. In addition, our mantissa truncation is specifically designed for a particular part of parameters in neural networks. We do not think this is related to GRIB2, a data description format specifically for weather data.
>
> > Weakness 1 (connection to non-uniform grid quantization): “Moreover, the paper fails to mention that this truncation itself is equivalent to quantisation to a non-uniform grid (f.e. float32 with 0 mantissa bits corresponds to quantization). Quantization with non-uniform grids is a widely used idea (f.e. in "Data Free Network Quantization", Chikin et al. 2022, or NF4 from QLoRA, Dettmers et al. 2023).”
>
> We would like to thank the reviewer for suggesting the previous paper "Data-free network compression via parametric non-uniform mixed precision quantization." by Chikin and Antiukh. However, we don’t find a straightforward connection between NeuZip and non-uniform grid quantization The non-uniform grid quantization essentially converts the original floating-point parameter into the fixed-point representation in a lossy manner. Our NeuZip, on the other hand, maintains the floating-point representation with up-to-lossless precision. Additionally, non-uniform grid quantization (including NF4 from QLoRA) assumes that the values to be compressed are centered at zero, whereas NeuZip could losslessly compress data centered around arbitrary values. Therefore, it is *false* that our method is a special case of non-uniform grid quantization.
>
> In fact, we deem the mentioned approach more similar to the NF4/FP4 data types from QLoRA (already cited and compared with in Section 3 of the initial submission). We have added a new section in the appendix (Appendix G) to discuss the connections and differences. We hope this response addresses the reviewer’s misunderstanding.
>
>
> > Weakness 2 (entropy coding not in baselines): “The methods that the paper compares lossless NeuZip to are not fairly chosen, and this is not immediately made obvious in the experimental descriptions. In Table 2, the quantisation methods that NeuZip compares to contain no entropy coding, while NeuZip contains entropy-coding in the exponent bits (which will be especially significant as the ratio of exponent bits to mantissa bits increases). Including entropy coding into the compared quantisation methods would reduce the speed by a small bit, but increase the compression performance at no perplexity cost (which are the metrics the methods are compared on). Again, as the proposed method is equivalent to non-uniform grid quantisation, I would expect most of the gains that are reported in this table to have come from the included entropy coding.”
>
> The reviewer complains that the baselines (i.e., previous studies) do not consider entropy coding. However, this is *not* a weakness of *our* work, but a weakness of previous work. We have compared state-of-the-art methods such as QLoRA in our work.
>
> In fact, entropy coding cannot be applied to previous work such as QLoRA and the recommended work  (non-uniform grid quantization). This is because the information-theoretically optimal quantization data types have an entropy equal to the number of bits, suggesting that they are not compressible anymore. This is also mentioned in both papers (QLoRA and non-uniform grid quantization).
>
> We have also included additional discussion in Appendix G.

---

> > ### Author Response · Authors · 2024-11-20
> > **Official Comment by Authors (2/2)**
> >
> > > Weakness 3 (LOMO is already good): “A large part of the memory efficiency gains of lossless NeuZip come from the established LOMO method (see Table 1). Adding NeuZip on top often reduces the execution speed by a large margin.”
> >
> > We acknowledge that LOMO is an effective memory-efficient training method and have properly cited and attributed the LOMO papers. However, we would like to point out that solely using LOMO consumes up to ~41% more memory without NeuZip, especially when the model is large. In addition, LOMO is a training-time approach, whereas NeuZip can be applied to both training and inference.
> >
> > > Question 1: I wonder how the reported speeds for the quantization methods such as Quant INT8 in Table 2 came to be. Especially for large models, inference speed-ups should be expected, see Dettmers et al. 2023 (LLM.int8(), https://arxiv.org/abs/2208.07339), appendix D, p.17ff, where LLM.int8 reports increased inference speed for all models that have more than 13B parameters).
> >
> > Thank you for this finding. However, the setting is different in the LLM.int8() paper: they tested the speedup on a feed-forward network only. In our paper, we report the end-to-end speed of each method. It is expected that LLM.int8() is still slower with 13B parameters because other smaller matrices (e.g. QKVO) could bottleneck the throughput. This result also aligns with [the observations](https://github.com/bitsandbytes-foundation/bitsandbytes/issues/6)  in the community.
> >
> > > Question 2: Similarly, for QLoRA, which has been shown to make networks more memory efficient in the original work, this paper seems to report contrary results, decreasing or barely reducing the memory efficiency compared to vanilla networks as well as suffering a large performance hit (f.e. T5 1B). A discussion of why this happens could be beneficial to the paper, and if it turns out that QLoRA is not suitable for networks this small, other state-of-the-art methods should rather be used to compare NeuZip with.
> >
> > Thank you for this comment. In the initial submission, we have discussed the potential reason in footnote 2. Essentially, quantization could lead to significant numerical instability when the weights have a wide dynamic range. In addition, QLoRA needs to add more parameters for tuning. By contrast, our training method is lossless and does not introduce additional parameters. We would also like to mention that, although our method performs well on small-scale settings, we do not intend to use them for justifications solely. This is because small models are usually not bottlenecked by hardware capabilities. In practical applications, the favorable scaling effect of NeuZip (demonstrated in the experiments) is more important.
> >
> >
> >
> > ---
> >
> > We thank the reviewer for providing the comments. The reviewer has misunderstandings about the novelty and baselines. In our response, we point out that:
> > * Our novelty does not lie in a new compression algorithm but lies in the information-theoretical observations that lead to an effective compression approach.
> > * The reviewer’s complaint about the lack of entropy coding in previous work is not a drawback of *our* work, and more importantly, entropy coding cannot be reasonably applied to previous work.
> >
> > We have added an appendix section to discuss these. We hope our response has addressed the misunderstandings and the reviewer could re-evaluate our paper. We are also happy to discuss any further concerns.

---

> > ### Comment · Reviewer_DiBQ · 2024-11-25
> >
> > > Our NeuZip, on the other hand, maintains the floating-point representation with up-to-lossless precision. Additionally, non-uniform grid quantization (including NF4 from QLoRA) assumes that the values to be compressed are centered at zero, whereas NeuZip could losslessly compress data centered around arbitrary values. Therefore, it is false that our method is a special case of non-uniform grid quantization.
> >
> > My comment specifically referred to *lossy* NeuZip, which includes truncation of the mantissa bits. This is **clearly** a type of non-uniform grid quantization, which can be easily seen in the special case I mentioned above: for 0 mantissa bits, we quantize to an exponentially spaced grid. For more mantissa bits, some points are included between the exponentially spaced grid points.
> >
> > The authors mention that NeuZip can losslessly compress data centered around arbitrary values, which is true, but again, my comment refers to lossy NeuZip specifically. For lossy NeuZip, the grid resulting from mantissa truncation is, by design (through the exponential spacing), also more sensitive for values closer to 0.
> >
> > > The reviewer complains that the baselines (i.e., previous studies) do not consider entropy coding. However, this is not a weakness of our work, but a weakness of previous work. We have compared state-of-the-art methods such as QLoRA in our work.
> > In fact, entropy coding cannot be applied to previous work such as QLoRA and the recommended work (non-uniform grid quantization). This is because the information-theoretically optimal quantization data types have an entropy equal to the number of bits, suggesting that they are not compressible anymore. This is also mentioned in both papers (QLoRA and non-uniform grid quantization).
> >
> > I would argue that using entropy coding is a very obvious step when trying to reduce the memory requirements of weights. The authors seem to agree with me, as they have also acknowledged that the novelty and contribution of their method lies in the identification of the parts of the weight vector to compress and the method of mantissa truncation. If their method does not compare favourably against adding a highly trivial entropy coding step to previous work, this calls into question the significance of their contribution.
> >
> > I therefore expect the authors to at least add entropy coding to the INT8 datatypes. Additionally, I would be interested in seeing a  demonstration of the authors’ claim that entropy coding the NF4 data types does not reduce the memory requirements at all. In Table 3 (b), the memory for NF4 is consistently higher than lossless (!) NeuZip. Assuming that the NF4 data type in this case is really incompressible, I would expect it to yield lower memory requirements (since it contains some lossy quantization of the original data) than lossless compression of the original weights. Maybe the authors can provide me with some clarity here.
> >
> > # Final Remark
> > Overall, the authors have clarified my doubts regarding the _lossless_ NeuZip experimental settings, and the motivation for the lossy NeuZip setting regarding the distinction between absolute and relative noise, as well as the connection to mantissa truncation.
> >
> > The authors clarified that the novelty of lossless NeuZip lies in the observation, which parts of the weight vector to compress. I maintain that this poses a pretty incremental contribution, as it should be quite clear that the mantissa bits (which vary on a much smaller scale) carry less information than the exponent bits, following from the well-known fact that neural networks are robust to some degree of noise and by the similarly well-known fact that neural network weights are mostly normal-distributed.
> >
> > Therefore, I maintain that the novelty of the paper remains a weak point, as all methods are well known and the observations that the authors make are not groundbreaking. Additionally, I doubt that the experiments properly show the effectiveness of the lossy NeuZip setting, as indicated in my comments above.

---

> > > ### Author Response · Authors · 2024-11-29
> > > **Official Comment by Authors (1/3)**
> > >
> > > We thank the reviewer for the timely response.
> > >
> > > > Reviewer’s response 1 (0-bit mantissa case = non-uniform quantization): “My comment specifically referred to lossy NeuZip, which includes truncation of the mantissa bits. This is clearly a type of non-uniform grid quantization, which can be easily seen in the special case I mentioned above: for 0 mantissa bits, we quantize to an exponentially spaced grid. For more mantissa bits, some points are included between the exponentially spaced grid points.”
> > >
> > > This is false. Non-uniform quantization and our method are different from each other. Specifically,
> > > * For 0-bit lossy NeuZip:
> > >     * Our method  represents values of $\pm2^{e_i}$, where $e_i$ is the exponent for the $i$th parameter. In the non-uniform quantization paper, the author suggested, the quantized values are $\pm \tau \frac{\sum_{j=0}^{k_i} 2^{j}}{\sum_{j=0}^{\tau-1} 2^{j}}$. By fundamental arithmetics, it is easy to see that they represent different values as the latter cannot be represented by the power of 2.
> > >     * Our method does not truncate exponent bits (indicating the range of $e_i$ above) in any cases, whereas the suggested paper controls the precision by tuning the range of $\tau$. Obviously, they do not work in the same way.
> > > * For NeuZip with more mantissa bits: they are even more different from the non-uniform quantization. In our method, each number has a unique exponent value, allowing every value to have its own magnitude. This also means that the values can have an arbitrary center. On the other hand, the suggested paper requires the values to have a center at zero, and all the values are scaled uniformly with a single magnitude. We also explained this in the initial response.
> > >
> > > Hypothetically, even if the 0-bit lossy NeuZip is the suggested non-uniform grid quantization (which, as explained above, is not the case), it would still underscore that our method represents a novel generalization of previous work, with 0-bit lossy NeuZip being a special case of NeuZip.
> > >
> > > We nevertheless included and discussed the non-uniform grid quantization in our paper as an example of quantization methods based on the suggestion. However, we do not agree with the reviewer’s criticism of our novelty.

---

> > > > ### Author Response · Authors · 2024-11-29
> > > > **Official Comment by Authors (2/3)**
> > > >
> > > > > Reviewers’s response 2 (suggesting using entropy coding is trivial) “I would argue that using entropy coding is a very obvious step when trying to reduce the memory requirements of weights. The authors seem to agree with me, as they have also acknowledged that the novelty and contribution of their method lies in the identification of the parts of the weight vector to compress and the method of mantissa truncation. If their method does not compare favourably against adding a highly trivial entropy coding step to previous work, this calls into question the significance of their contribution.”
> > > >
> > > > We do not agree with the comment that applying entropy coding is an obvious step. We have explained, in the initial submission and the added appendix, that (1) our observation in exponent entropy is an essential foundation to enable efficient entropy encoding; and (2) previous quantization methods are not suitable for entropy encoding. See the response below for evidence.
> > > >
> > > > >  Reviewers’s response 2 (suggesting conducting experiments where we add entropy coding to baselines) “I therefore expect the authors to at least add entropy coding to the INT8 datatypes. Additionally, I would be interested in seeing a demonstration of the authors’ claim that entropy coding the NF4 data types does not reduce the memory requirements at all.”
> > > >
> > > > We are confident that our claim is correct and conduct experiments to show the entropy of quantized values. Specifically, we examined the quantized parameters and calculated their entropy values. The results are as follows:
> > > >
> > > > |Model+Type|Entropy|Ent/#bits (%)|Maximum reduction| Distribution|
> > > > |---|---|---|---|---|
> > > > |Llama 3.2 1B + INT8|7.05|88.1%|0.11GiB|[Figure](https://anonymous.4open.science/api/repo/NeuZip-Submission-FDED/file/notebooks/meta-llama_Llama-3.2-1B_int8.png?v=8146348f)|
> > > > |Llama 3.2 1B + FP4|3.70|92.6%|0.034GiB|[Figure](https://anonymous.4open.science/api/repo/NeuZip-Submission-FDED/file/notebooks/meta-llama_Llama-3.2-1B_fp4.png?v=6f9916b0)|
> > > > |Llama 3.2 1B + NF4|3.87|96.7%|0.015GiB|[Figure](https://anonymous.4open.science/api/repo/NeuZip-Submission-FDED/file/notebooks/meta-llama_Llama-3.2-1B_nf4.png?v=c89564dc)|
> > > > |Llama 3 8B + INT8|7.01|87.6%|0.81GiB|[Figure](https://anonymous.4open.science/api/repo/NeuZip-Submission-FDED/file/notebooks/meta-llama_Meta-Llama-3-8B_int8.png?v=1db0f225)|
> > > > |Llama 3 8B + FP4|3.71|92.7%|0.24GiB|[Figure](https://anonymous.4open.science/api/repo/NeuZip-Submission-FDED/file/notebooks/meta-llama_Meta-Llama-3-8B_fp4.png?v=9cb8031d)|
> > > > |Llama 3 8B + NF4|3.87|96.9%|0.1GiB|[Figure](https://anonymous.4open.science/api/repo/NeuZip-Submission-FDED/file/notebooks/meta-llama_Meta-Llama-3-8B_nf4.png?v=16c442a5)|
> > > >
> > > > As shown, after quantization, the quantized values are not easily compressible. Even the linear (uniform) quantization with INT8 can only be reduced by 0.81 GiB maximum for Llama 3 8B. In Table 3, INT8 uses 0.93 GiB more than NeuZip 3-bit while underperforms NeuZip in both perplexity and speed. In addition,  the best-performing 4-bit data type, NF4, can only be deflated by 3.3% maximum, aligned with our prediction that optimal quantization is not compressible. In addition, adding another layer of compression will halve the throughput, making quantization methods even slower. This provides a demonstration (as requested by the reviewer) for our claim in the paper that optimally quantized values are not easily compressible (lines 982-986).
> > > >
> > > > Given the results, we show that the reviewer’s suggested approach is infeasible and our comparisons are fair. We hope our empirical verification serves as a strong confirmation of our claim.
> > > >
> > > > To fully convince the reviewer of our analysis, we have uploaded the Jupyer notebook to our anonymous code repository. The reviewer can directly check our implementation and easily replicate our results by one click. Note that all of the code and figures do not contain any authors’ information. We are also willing to offer technical support if the reviewer encounters any issues.

---

> > > > > ### Author Response · Authors · 2024-11-29
> > > > > **Official Comment by Authors (3/3)**
> > > > >
> > > > > >  Reviewers' response 2 (doubts for Table 3 (b)): “ In Table 3 (b), the memory for NF4 is consistently higher than lossless (!) NeuZip. Assuming that the NF4 data type in this case is really incompressible, I would expect it to yield lower memory requirements (since it contains some lossy quantization of the original data) than lossless compression of the original weights. Maybe the authors can provide me with some clarity here.”
> > > > >
> > > > > As explained, the lossy nature of quantization methods can be detrimental to some models. Therefore, certain parameters are required to maintain the original precision. Otherwise, the loss is severely exploded. As a result, the excessive memory usage is not due to the use of NF4 or other quantization types, but due to the inability to apply them because of their low precision. This standard pipeline is already mentioned in the initial submission and in the response. We refer to [this community discussion thread](https://github.com/huggingface/transformers/issues/20287) for more details. In addition, our anonymous code repository contains all the files needed to replicate our experiments (including Table 3). We welcome the reviewer to check and run the `examples/inference/s2s_perplexity.py` file provided in our [anonymous repository](https://anonymous.4open.science/r/NeuZip-Submission-FDED) for replication. We hope this clarifies our results.
> > > > >
> > > > >
> > > > > ### **Final remark**
> > > > >
> > > > > We sincerely thank the reviewer for acknowledging our response addressing the concerns. The reviewer mainly thinks our novelty is weak and observations are not groundbreaking. However, we would like to point out that NeuZip is a substantially different method and demonstrates favorable and unique properties compared with previous methods (including the one that the reviewer additionally suggested). Our claims are fully supported by detailed analyses and empirical verifications.
> > > > >
> > > > > Furthermore, we particularly want to point out that the criterion of “groundbreaking” is subjective and unreasonable for evaluating a submission to ICLR, which accommodates thousands of papers; thus, such a review is unfair to us. We believe our research offers novel and unique insights and would be a net gain for the community. This is commonly recognized as a proper contribution as stated in the [ICLR review guidelines](https://iclr.cc/Conferences/2025/ReviewerGuide): “Submissions bring value to the ICLR community when they convincingly demonstrate new, relevant, impactful knowledge (incl., empirical, theoretical, for practitioners, etc)”.

---

> > > > ### Comment · Reviewer_DiBQ · 2024-12-02
> > > > **Clarification Regarding Author Comment 1**
> > > >
> > > > > Author Comment 1 (Non-Uniform Grid)
> > > >
> > > > As I see now, there must be a fundamental misunderstanding about my comments. My initial comment did not refer to the specific paper by Chikin et al., nor did I refer to any paper in particular, but to the general concept of quantizing to a non-uniform grid, which is a widely used idea (but of course, the exact design of the grid varies). Again, I maintain that lossy NeuZip is **an instance of** non-uniform quantization.

---

> ### Author Response · Authors · 2024-12-03
>
> Dear reviewer. We sincerely appreciate your reply. However, we do not think this criticism is valid.
>
> 1. We have discussed and explained in both responses why NeuZip is **not** an instance of non-uniform quantization. In particular, we showed that it is a generalization of quantization methods. It also has more favorable properties (e.g. high precision up to lossless, applicable to training, etc) compared with quantization. We hope the reviewer can thoroughly read our responses.
> 2. We would like to remind the reviewer that by this logic, all quantization methods (not including NeuZip) are grid quantization with their own grid designs. If this criticism holds, it essentially discards the progress of the quantization methods in deep learning and is not particularly about our method.
>
> Beyond this specific criticism, the reviewer originally had doubts about our experiment settings, so we provided more information, such as theoretical justification and empirical evidence, to show our experiments were fair. We also uploaded the relevant code to the anonymous repo for easy replication. We thank the reviewer for the suggestions and believe the concerns are addressed. Should the reviewer have any comments that lead to a negative score, please do not hesitate to let us know before the deadline.

---

### Official Review · Reviewer_UCfs · 2024-11-03

**Soundness:** 4
**Presentation:** 3
**Contribution:** 4
**Rating:** 8
**Confidence:** 3

**Summary:**

This is an interesting work that uses small entropy in the exponent bit to compress neural networks in two settings: lossless and lossy. In the lossless setting the memory consumption and inference speed are both improved. The authors try their models on LLMs with <70B parameters but the method is general and can be used to compress LLMs of larger size. The paper is properly structured, well-written, and easy to follow.

**Strengths:**

The paper addresses the important problem of increased size of LLMs with lack of resources for training and inference. The algorithm can be built on top of other compression methods. The authors also show that their lossy compression falls on the Pareto frontier and therefore a good compression candidate overall.

**Weaknesses:**

The paper motivates on linear layer, and their compression requires activations to be saved. This is a restriction specially when small values like GELU type activations are generated in the network. I wonder if the paper suits partial linear activations like ReLU better. Discussion on convutional models, and RNNs will strengthen the paper further.

**Questions:**

- Transformer-type language models such as BERT-type are getting implemented on edge devices such as cell phones. Do you see a potential that this compression helps the edge implementation of LLMs?
- Do you see a potential for Transformer-alternative architectures such as MAMBA, xLSTMs, JAMBA, etc?

---

> ### Author Response · Authors · 2024-11-20
>
> Thank you for your insightful comments and your strong support.
>
> > The paper motivates on linear layer, and their compression requires activations to be saved. This is a restriction specially when small values like GELU type activations are generated in the network. I wonder if the paper suits partial linear activations like ReLU better. Discussion on convutional models, and RNNs will strengthen the paper further.
>
> Thanks for the suggestion! We are adding Appendix F to discuss the application of our work to other architectures.
>
> * In particular, CNN and RNN (other than the non-linear activation function)can be viewed as extensions to linear transformation. Therefore, our approach is applicable.
> * Activation functions like GELU may demand more memory. However, this applies to not only NeuZip but also to all other training pipelines. We consider lossless compression of activations as a potential solution in future work.
>
> In our original submission, we focused on Transformer models due to their popularity. Based on the comment (thanks again!), we included an additional experiment with Mamba (which is related to both CNN and RNN) in Appendix F. Results show that NeuZip provides consistent memory saving without compromising the model performance and speed.
>
>
> > Transformer-type language models such as BERT-type are getting implemented on edge devices such as cell phones. Do you see a potential that this compression helps the edge implementation of LLMs?
>
> Yes, definitely. We consider edge devices an ideal use case given that the memory is usually limited. We have added this in future work.
>
> > Do you see a potential for Transformer-alternative architectures such as MAMBA, xLSTMs, JAMBA, etc?
>
> Yes! Following the suggestion, we conducted additional experiments with the Mamba models (Appendix F). See details in the first point of this response. Thank you!
>
> ---
>
> We are especially grateful to the reviewer for recognizing the novelty and effectiveness of our method. We have included the related discussion in the paper to further strengthen our work. Please let us know if you have any questions.

---

> > ### Comment · Reviewer_UCfs · 2024-11-22
> > **Satisfied with the response**
> >
> > Thank you. I am satisfied with the reply, and I keep my score.

---

> > > ### Author Response · Authors · 2024-11-29
> > >
> > > Thank you for confirming that our response is satisfactory. We appreciate your continued positive view of our paper and the time you have dedicated to the review. We are pleased to have your support.

---

### Official Review · Reviewer_nr8v · 2024-11-04

**Soundness:** 2
**Presentation:** 1
**Contribution:** 1
**Rating:** 3
**Confidence:** 2

**Summary:**

This paper presents a entropy based quantization method to reduce the memory footprint in training. It is claimed that this method could reduce around half of the memory without losing accuracy.

**Strengths:**

The paper made relatively extensive experiments on the modern LLMs, and showed its advantage in reducing memory footprint.

**Weaknesses:**

1. The motivation is unclear. It's not clear that where the entropy for the sign bit (Figure 1) comes from. The author did not mention in details, where the pre-trained model weight is obtained and whether it is an empirical value or theoretical value, making the analysis not persuasive enough.

2. The compression algorithm is not clearly presented.

3. Codes are not released, otherwise we can make clear what the methodology is by checking the implementation.

**Questions:**

Are the experiments of pre-trained being ran for a full epoch? How many A6000s are used and how long it took?

---

> ### Author Response · Authors · 2024-11-20
>
> We thank the reviewer for asking questions related to our work.
>
>
> > The motivation is unclear. It's not clear that where the entropy for the sign bit (Figure 1) comes from. The author did not mention in details, where the pre-trained model weight is obtained and whether it is an empirical value or theoretical value, making the analysis not persuasive enough.
>
>
> As mentioned in lines 130-137, we stated our motivation as “the number of bits per exponent is largely inflated compared with the information entropy”, and “therefore propose to compress the exponent bits”. We also visualize this motivation in Figure 1 for a clear display of the observation.
>
> As mentioned in lines 98-99, the sign bit comes from the floating-point representation given by the IEEE 754 standard. This is also visualized in our Figure 8.
>
> As mentioned in the caption of Figure 1, the statistics come from the Llama-3 8B model. We also provided the information on where we obtained the weights in the “Reproducibility Statement” section.
>
> > The compression algorithm is not clearly presented.
>
> Thank you for the comment. We only briefly mentioned the compression algorithm because (1) our method is agnostic to the underlying compression algorithms and (2) we have already reached the 10-page limit. For completeness, we did outline how it achieves entropy-based compression and properly cited previous work in lines 190-196. The purpose of this paper is not to reinvent a general compression algorithm. Instead, we focus on identifying and exploiting the compressibility of neural networks.
>
> > Codes are not released, otherwise we can make clear what the methodology is by checking the implementation.
>
> As stated in line 569, we included the anonymous code in the “Reproducibility Statement” section.
>
> > Are the experiments of pre-trained being ran for a full epoch? How many A6000s are used and how long it took?
>
> Yes. As stated in lines 262-264, we choose to run one epoch because our method for training is fully lossless. Running more epochs would not help strengthen our empirical results.
>
> We use one A6000 for each experiment to avoid complications with the memory usage in the multi-GPU scenario. We have updated the PDF to make this clearer (positioned above Sec 3.1). The scales of the absolute run time under different settings are quite different and grow with iteration steps. Instead, we reported the speed metric in iterations per second in all experiments.
>
> ---
>
> Overall, we thank the reviewer for asking questions related to our work. As shown in our response, most of the questions were already answered in the original submission.  Nevertheless, we have made revisions to further clarify. We urge the reviewer to thoroughly read our paper again and  re-evaluate it accordingly. Please do not hesitate to ask further questions. Thanks!

---

> ### Author Response · Authors · 2024-11-29
>
> Dear reviewer, we haven’t received your further comments since our response more than one week ago. Since the discussion deadline is approaching, we would like to know if your questions still remain and consequently warrant a negative score. In any case, we are fully committed to answering all the questions you might have.

---

### Author Response · Authors · 2024-11-20
**Updated PDF**

Dear all reviewers,

We sincerely thank you for your efforts and insightful comments. We have updated the PDF file to reflect our responses to the discussion. All of our edits are marked in orange to distinguish them from the initial version. Specifically, we mainly made the following changes:

* Added the statement on the hardware for each experiment to answer Reviewer nr8v’s question.
* Added the discussion in Appendix F for the cases beyond linear layers based on Reviewer UCfs’s suggestion. We also added an experiment based on Mamba models to support our main claims and verify the effectiveness of NeuZip in broader applications.
* Added the discussion on our compression versus previous papers based on quantization in Appendix G according to Reviewer DiBQ’s comments.

We hope our responses clear out the misunderstandings in the initial reviews. Please feel free to raise any additional questions.

Best regards, \
The Authors

---

### Comment · Area_Chair_x5f4 · 2024-11-22
**Discussion**

Dear reviewers,

The authors have responded to your reviews.

Until November 26th @ 2359 (AOE time) reviewers and authors can freely exchange responses, so if there any clarifications you require from the authors, now is the time to seek them!

Best,

AC

---

### Meta-Review · Area_Chair_x5f4 · 2024-12-17

**Metareview:**

This paper proposes NeuZip, a weight compression technique that takes account of the entropy of floating point numbers in NNs. The authors propose a lossless, and lossy variant, and evaluate these on various language models.

I will not be taking Reviewer nr8v's review into account for this meta review, as it is extremely short and poor quality. I asked the reviewer the day after the review was posted to provide a high quality review ant this was ignored.

In terms of strengths, Reviewer UCfs was appreciative of the problem tackled, and the experimental results. They also approved of how the algorithm could be combined with existing techniques. Reviewer DiBQ thought the paper was strong on presentation.

In terms of weaknesses, Reviewer UCfs wanted to see applicability to model types beyond transformers (which I note that the authors provide with Mamba experiments). Reviewer DiBQ had major concerns about novelty, and the experimental setup.

After an extensive discussion, Reviewer DiBQ still has concerns that this work does not provide much additional knowledge to the community, and that the experimental setup presented is unfairly chosen to benefit the proposed method. From my perspective, the negatives presented outweigh the positives (it isn't clear what elevates this work from the bulk of existing compression methods), so I am leaning towards rejection. However, I am not fully confident in my decision as I am effectively basing this on 2 reviews, so I will indicate this in my confidence score.

**Additional Comments On Reviewer Discussion:**

Reviewer nr8v didn't engage with the review process, which was a shame. Reviewer UCfs's only real criticism was a lack of experiments on other network archetypes, which the authors provided, causing to the reviewer to stick with their initial score. There was an extensive back-and-forth between the authors and Reviewer DiBQ which seemed to hinge on a misunderstanding of a comment about quantisation grids. The reviewer indicated that this didn't really influence their score, so I have not taking this into account in my decision. Their main issues about novelty and experimental setup were not satisfactorily addressed which had a significant weight on my final decision.

---

### Decision · Program_Chairs · 2025-01-22

Reject